# GENERATING SYMBOLIC REASONING PROBLEMS WITH TRANSFORMER GANS

## ABSTRACT

Constructing training data for symbolic reasoning domains is challenging: Existing instances are typically hand-crafted and too few to be trained on directly and synthetically generated instances are often hard to evaluate in terms of their meaningfulness. We study the capabilities of GANs and Wasserstein GANs equipped with Transformer encoders to generate sensible and challenging training data for symbolic reasoning domains. We conduct experiments on two problem domains where Transformers have been successfully applied recently: symbolic mathematics and temporal specifications in verification. Even without autoregression, our GAN models produce syntactically correct instances. We show that the generated data can be used as a substitute for real training data when training a classifier, and, especially, that training data can be generated from a real dataset that is too small to be trained on directly. Using a GAN setting also allows us to alter the target distribution: We show that by adding a classifier uncertainty part to the generator objective, we obtain a dataset that is even harder to solve for a classifier than our original dataset.

## 1 INTRODUCTION

Deep learning is increasingly applied to more untraditional domains that involve complex symbolic reasoning. Examples include the application of deep neural network architectures to SAT (Selsam et al., 2019; Selsam & Bjørner, 2019; Ozolins et al., 2021), SMT (Balunovic et al., 2018), temporal specifications in verification (Hahn et al., 2021; Schmitt et al., 2021), symbolic mathematics (Lample & Charton, 2020), or theorem proving (Loos et al., 2017; Bansal et al., 2019; Huang et al., 2019; Urban & Jakubuv, 2020).

The acquisition of training data for symbolic reasoning domains, however, is a challenge. Existing instances, such as benchmarks in competitions (Biere & Claessen, 2010; Froleyks et al., 2021; Jacobs et al., 2017) are typically hand-crafted, for example, in a "bring your own benchmarks" setting (Balyo et al., 2017). Since the instances are too few to be trained on, training data is, thus, typically generated synthetically. For example by random sampling (Selsam et al., 2019; Lample & Charton, 2020), or by randomly re-combining parts of existing instances (Schmitt et al., 2021). Although these data generation methods already lead to good results, training on randomly generated data carries the risk of training on meaningless data or the risk of introducing unwanted biases.

In this paper, we study the generation of symbolic reasoning problems with Generative Adversarial Networks (GANs) (Goodfellow et al., 2014) and show that they can be used to construct large amounts of meaningful training data from a significantly smaller data source. GANs, however, can not immediately be applied: Symbolic reasoning problems reside typically in a discontinuous domain and, additionally, training data is typically sequential and of variable length. We show that training directly in the one-hot encoding space is possible when adding Gaussian noise to each position. We, furthermore, use a Transformer (Vaswani et al., 2017) encoder to cope with the sequential form of the data and the variable length of the problem instances.

We provide experiments to show the usefulness of a GAN approach for the generation of reasoning problems. The experiments are based around two symbolic reasoning domains where recent studies on the applicability of deep learning relied on large amounts of artificially generated data: symbolic mathematics and linear-time temporal logic (LTL) specifications in verification. We report our experimental results in three sections. We first provide details on how to achieve a stable training of

a standard GAN and a Wasserstein GAN (Arjovsky et al., 2017) both equipped with Transformer encoders. We analyze the particularities of their training behavior, such as the effects of adding different amounts of noise to the one hot embeddings. Secondly, we show for an LTL satisfiability classifier that the generated data can be used as a substitute for real training data, and, especially, that training data can be generated from a real dataset that is too small to be trained on directly. In particular, we show that out of 10K real training instances, a dataset consisting of 400K instances can be generated, on which a classifier can successfully be trained on. Lastly, we show that generating symbolic reasoning problems in a GAN setting has a specialty: We can alter the target distribution by adding a classifier uncertainty part to the generator objective. By doing this, we show that we can obtain a dataset that is even harder to solve than the original dataset which has been used to generate the data from.

The remainder of this paper is structured as follows. In Section 2, we give a short introduction to the problem domains considered in this paper and describe how the origin training data has been constructed. In Section 3, we present our Transformer GAN architecture(s), before providing experimental results in Section 4. We give an overview over related work in Section 5 before concluding in Section 6.

## 2 PROBLEM DOMAIN AND BASE DATASETS

In this section, we introduce the two problem domains on which we base our experiments on: satisfiability of temporal specifications for formal verification and function integration and ordinary differential equations (ODEs) for symbolic mathematics. We furthermore give an overview over the data generation processes of these base datasets.

### 2.1 HARDWARE SPECIFICATIONS IN LINEAR-TIME TEMPORAL LOGIC (LTL)

Linear-time Temporal Logic (LTL) (Pnueli, 1977) is the basis for industrial hardware specification languages like the IEEE standard PSL (IEEE-Commission et al., 2005). It is an extension of propositional logic with temporal modalities, such as the Next-operator ($\bigcirc$) and the Until-operator ($\mathcal{U}$). There also exist derived operators, such as "eventually" $\Diamond \varphi$ ($\equiv true\,\mathcal{U}\,\varphi$) and "globally" $\Box \varphi$ ($\equiv \neg \Diamond \neg \varphi$). For example, mutual exclusion can be expressed as the following specification: ($\neg \Box (access_{p0} \wedge access_{p1})$), stating that processes $p0$ and $p1$ should have no access to a shared resource at the same time. The base problem of any logic is its satisfiability problem. It is the problem to decide whether there exists a solution to a given formula. The satisfiability problem of LTL is a hard problem, in fact, it is PSPACE-hard Sistla & Clarke (1982). The full syntax, semantics and additional information on the satisfiability problem can be found in Appendix A.

So far, the construction of datasets for LTL formulas has been done in two ways (Hahn et al., 2021): Either by obtaining LTL formulas from a fully random generation process, which likely results in unrealistic formulas, or by sampling conjunctions of LTL specification patterns (Dwyer et al., 1999). To obtain a healthy amount of unsatisfiable and satisfiable instances in this artificial generation process, we slightly refined the pattern-based generation method with two operations. Details can be found in Appendix B. Since the formula length correlates to unsatisfiability, we filter for equal proportions of classes per formula length. We restrict the tree size of the formulas to 50. We call this dataset `LTLbase`.

### 2.2 SYMBOLIC MATHEMATICS

Lample & Charton (2020) showed that Transformer models perform surprisingly well on symbolic mathematics. More precisely, they applied the models to function integration and ordinary differential equations (ODEs).

We consider the function integration problem and use the forward generated dataset (`https://github.com/facebookresearch/SymbolicMathematics`). Random functions with up to $n$ operators are generated and their integrals are calculated with computer algebra systems. Functions that the system cannot integrate are discarded. Mathematical expressions are generated randomly. The dataset is cleaned, with equation simplification, coefficients simplification, and filtering out invalid expressions (Lample & Charton, 2020). We restrict the tree size to 50.

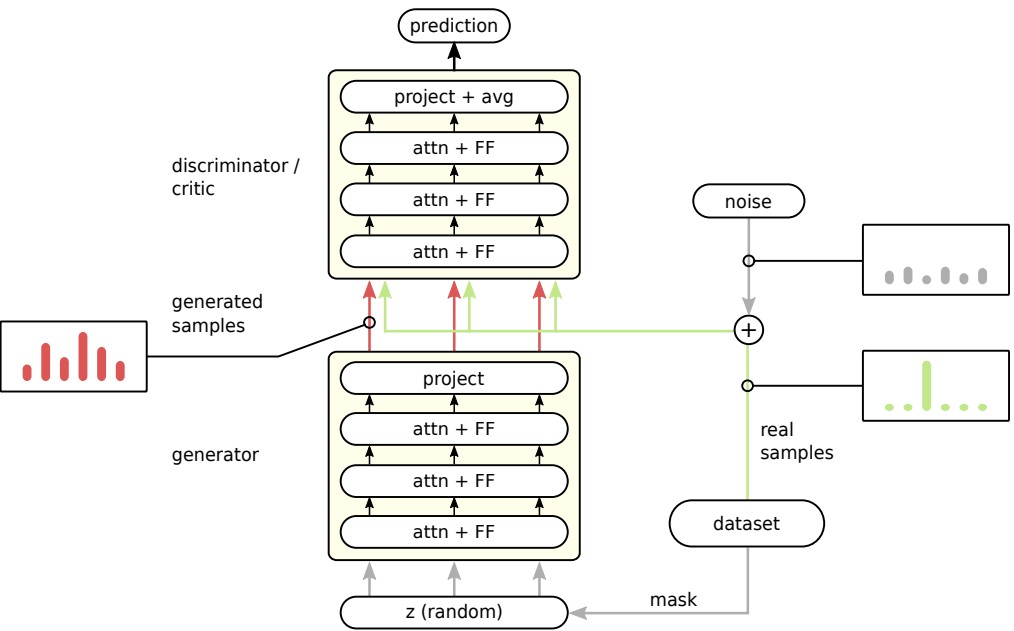

Figure 1: TGAN-SR: Transformer GAN for generating symbolic reasoning problems with visualizations of the per-position one-hot space.

## 3 ARCHITECTURE

The Transformer GAN architecture for generating symbolic reasoning problems (TGAN-SR) is depicted in Figure 1. It consists of two Transformer encoders as discriminator/critic and generator, respectively. The inner layers of the encoders are largely identical to standard transformers (Vaswani et al., 2017), but their input and output processing is adjusted to the GAN setting. We use an embedding dimension of $d_{emb} = 128$, $n_h = 8$ attention heads, and a feed-forward network dimension of $d_{FF} = 1024$ for both encoders as default.

The generator's input is a real scalar random value with uniform distribution $[0, 1]$ for each position in the sequence. It is mapped to $d_{emb}$ by an affine transformation before being processed by the first layer. The position-wise padding mask is copied from the real data during training, so the lengths of real and generated formulas at the same position in a batch are always identical. During inference, the lengths can either be sampled randomly or copied from an existing dataset similar to training. Either way, the generator encoder's padding mask is predetermined so it has to adequately populate the unmasked positions. With $V$ being the vocabulary, and $|V|$ being the size of the vocabulary, an affine transformation to dimensionality $|V|$ and a softmax is applied after the last layer. The generator's output lies, thus, in the same space as one-hot encoded tokens. We use $n_{lG} = 6$ layers for our default model's generator.

A GAN discriminator and WGAN critic are virtually identical in terms of their architecture. The only difference is that a critic outputs a real scalar value where a discriminator is limited to the range $[0, 1]$, which we achieve by applying an additional logistic sigmoid in the end. To honor their differences regarding the training scheme, we use both terms when referring to exchangeable properties and make no further distinctions between them. For input processing, their $|V|$-dimensional (per position) input is mapped to $d_{emb}$ by an affine transformation. After the last layer, the final embeddings are aggregated over the sequence by averaging and a linear projection to a scalar value (the prediction logit) is applied. Our default model uses $n_{lD} = 4$ layers. We achieved best results with slightly more generator than discriminator/critic layers. A full hyperparameter study can be found in AppendixC.2.

Working in the $|V|$-sized one-hot domain poses harsh constraints on the generator's output. Contrary to continuous domains were GANs are usually employed, each component of a real one-hot vector is, by definition, either 0 or 1. If the generator were to identify this distribution and use it as criterion to

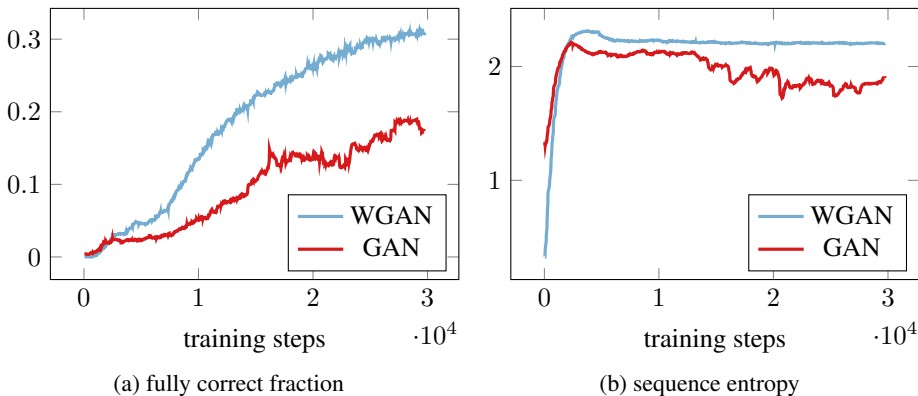

Figure 2: Quality measures for the GAN and WGAN variant when generating temporal specifications.

tell real and generated instances apart, this would pose a serious difficulty for training. We therefore sample a $|V|$-sized vector of Gaussian noise $N(0, \sigma_{\text{real}}^2)$ for each position (see Figure 1). We add it to the real samples' one-hot encoding and re-normalize it to sum 1 before handing them to the discriminator/critic. By default, we use a value of $\sigma_{\text{real}} = 0.1$ for all models to get comparable results. We study the effect of different amounts of noise more closely in Section 4.1.2.

## 4 EXPERIMENTS

In this section, we report our experimental findings. We structure our results in three sections. We first report on the performance of the TGAN-SR architecture in constructing syntactically correct instances of temporal specifications and mathematical expressions. Secondly, we show, exemplary for LTL formulas, that the newly generated dataset can be used as a substitute for the origin dataset. Lastly, we show, by altering the target distribution, that the network can generate a dataset that is harder to solve for a classifier. We trained the models on an NVIDIA DGX A100 system for around 8 hours. We begin each subsection with a short preamble on the training setting.

### 4.1 PRODUCING SYNTACTICALLY CORRECT SYMBOLIC REASONING PROBLEMS

The goal of the experiments in this section is to asses the generator's capability in creating valid symbolic reasoning problems as objectively as possible. If not stated otherwise, in plots and tables, we report results from our default model averaged across three runs and with an exponential smoothing ($\alpha = 0.95$) applied. For temporal specifications, we use LTLbase as training set and for symbolic math the dataset described in section 2.2.

#### 4.1.1 TRAINING SETTING

For the GAN variant, we employ the standard GAN training algorithm (Goodfellow et al., 2014). For our default model, we use $n_c = 2$ discriminator training steps per generator training step and a batch size of $bs = 1024$. Notably, we use the alternative generator loss $-\mathbb{E}_{z \sim p_z}\left[\log D(G(z))\right]$ instead of the theoretically more sound $\mathbb{E}_{z \sim p_z}\left[\log\left(1 - D(G(z))\right)\right]$. The WGAN variant uses the WGAN-GP training with gradient penalty as proposed by Gulrajani et al. (2017) with $\lambda_{GP} = 10$. Standard WGAN losses are used and the training loop parameters $n_c$ and $bs$ are identical to the GAN variant. To calculate the gradient penalty of intermediate data points according to Gulrajani et al. (2017), we make use of the fact that for each batch element, real and generated samples share the padding mask. After the gradient with respect to an intermediate point is calculated, the gradient's squared components are masked out at padded positions before being summed up over the sequence length. For both variants, both discriminator and generator are trained with the Adam (Kingma & Ba, 2015) optimizer ($\beta_1 = 0, \beta_2 = 0.9$) and constant learning rate $lr = 1e - 4$, similar to Gulrajani et al. (2017).

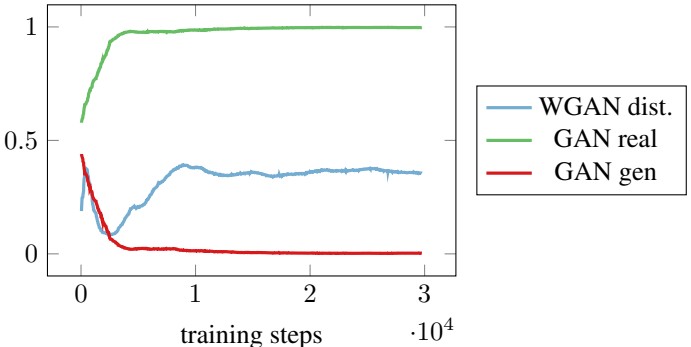

Figure 3: GAN real/generated predictions and WGAN Wasserstein distance estimate when generating temporal specifications.

### 4.1.2 RESULTS

**Generating valid symbolic reasoning problems.** During training we periodically sample several generated instances and convert them to their text representation, which involves taking the *argmax* at every position. We then try to parse a prefix-encoded tree from the resulting tokens. If the parsing of a problem is successful and no tokens remain in the sequence, we note this problem as *fully correct*. The fraction over the course of training to generate temporal specifications is depicted in Figure 2a. Both GAN and WGAN variants increase the measure relatively continuously, but eventually reach their limit around 30K training steps. Still, both generators are able to produce a large fraction of fully correct temporal specifications, despite the length of the instances (up to 50 tokens) and the non-autoregressive, fixed-step architecture. We list some examples below:

$$\neg(h \to h) \, \mathcal{W} \bigcirc (g \vee h) \wedge (g \wedge g) \wedge \Diamond \Box \neg \Box j \wedge \neg \Box j \, \mathcal{W} \neg b \wedge \Box (\Box h \wedge \bigcirc j \to \Box j) \wedge \Diamond \Diamond j \quad,$$
$$\bigcirc \bigcirc \bigcirc (c \vee i) \wedge \neg d \wedge \neg \Diamond \bigcirc c \, \mathcal{W} \neg c \wedge \Box (\Box d \wedge \neg ((b \leftrightarrow c) \leftrightarrow$$
$$\leftrightarrow \Box c) \to \bigcirc (c \leftrightarrow \Box d)) \wedge \Box (b \wedge d \to \Diamond \Diamond \Box \Box d) \wedge c \quad.$$

The network also produces correct symbolic mathematical expressions when training on the forward generated mathematical dataset of Lample & Charton (2020). After 30K steps, on average 30% are fully correct. We list some examples below:

$$x^3 \cdot ((-1) \cdot (\ln x)^3 + 2 \cdot x \cdot (\mathrm{acosh}\,(5) + 1 + (-1) \cdot x \cdot (2 + x)^{44})) \quad,$$
$$1 \div 2 \cdot 81264 \div x \cdot 1 \div 5 \cdot x \cdot \ln 4 + 2 \quad,$$
$$x \cdot (3 \cdot (x^3) + x \cdot 2) + (2 + x) \cdot 4 \cdot x \cdot (1 \div 201 + \mathrm{acos}(44)) \quad.$$

**Differences in homogeneity.** Comparing the valid generated formulas from the WGAN and GAN variants, we find that often, the latter would produces formulas in the likes of

$$\bigcirc \bigcirc \bigcirc \bigcirc \bigcirc \bigcirc \bigcirc \bigcirc \bigcirc \bigcirc \bigcirc \Diamond i \wedge \bigcirc \bigcirc i \wedge \bigcirc \bigcirc \bigcirc \bigcirc \neg \bigcirc \Box \neg \neg \Diamond i \wedge \neg \neg (g \wedge g \wedge i) \quad \text{or}$$
$$\tanh 12225556676677799655766669 \cdot x \quad,$$

which contains repetitions (of the $\bigcirc$-operator) or easily stringed together sequences (for example of numbers). In fact, some GAN runs achieved fully correct fractions above 30% (higher than WGAN), but these exclusively produced formulas with such low internal variety. To quantify this, we calculated a *sequence entropy* which treats the number of occurrences of the same token in the sequence relative to the sequences length as probability. Figure 2b shows that indeed this metric decreases for the GAN variant during training but remains stable for WGANs. We therefore speculate that the discriminator/critic indeed learns to check syntactic validity to some extend and some generators "exploit" this fact by producing correct, but repetitive formulas. For further experiments that use generated instances, we therefore exclusively stick to the WGAN variant.

**Discriminator / critic predictions.** We observe a quick identification of real and generated instances by the GAN discriminator as depicted in Figure 3. Predictions reach values above 0.99 and

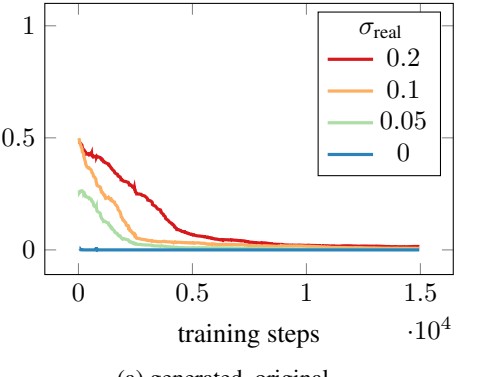
(a) generated, original

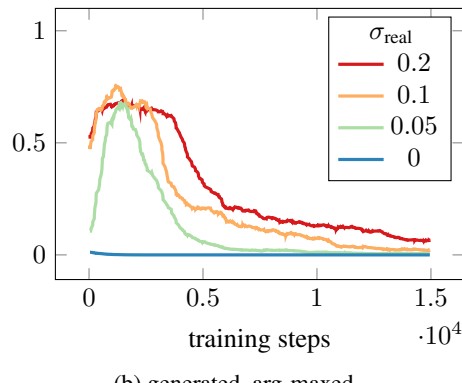
(b) generated, arg-maxed

Figure 4: GAN discriminator predictions for generated samples with different noise level $\sigma_{\text{real}}$ on real samples when generating temporal specifications.

below 0.01, respectively, and never change directions. Similarly, the WGAN critic's Wasserstein distance estimate soon reaches a value of around 0.4 at which it remains for the rest of training. For this behavior, one would expect the generator to not improve significantly, which is contrary to the observed improvements in quality.

**Effects of additive noise on one-hot representation.** We also studied the effect of adding different amounts of noise to the one-hot representation of real temporal specification instances (see Table 1). It strongly affects the performance of the GAN scheme, which is unable to work without added noise. Stronger noise however improves this variants performance. WGAN models on the other hand were not significantly influenced by added noise and are able to be trained without it.

Additionally, we compare how the GAN discriminator rates unmodified generated instances and *argmaxed* versions thereof (see Figure 4). For this, we also evaluate *argmaxed* instances during each step of training without changing the training regime. While the score for unmodified instances immediately decreases at the start of the training, it initially rises for the *argmaxed* ones. After a while of training, though, the scores of the *argmaxed* samples quickly deteriorate and, at least for lower values of $\sigma_{\text{real}}$, approach their soft-valued counterparts. A possible interpretation is that the discriminator first identifies generated samples by their different distributions in the one-hot domain, which, naturally, is eased with low noise on the real samples, before shifting its focus from this low-level criterion to more global features.

## 4.2 Substituting Training Data with Generated Instances

In this subsection, we show that the origin training data can be substituted with generated training data when training a classifier on the LTL satisfiability problem.

Table 1: Comparison on fully correct formulas ($fc$) and sequence entropy ($se$) of GAN and WGAN with different $\sigma_{real}$ when generating temporal specifications. 3-run average, smoothed ($\alpha = 0.95$), standard deviations in Table 5.

| architecture | $\sigma_{real}$ | $fc$ | $se$ | architecture | $\sigma_{real}$ | $fc$ | $se$ |
|---|---|---|---|---|---|---|---|
|  | 0 | 0% | - |  | 0 | 26% | 2.2 |
|  | 0.05 | 15% | 1.7 |  | 0.05 | 25% | 2.2 |
| GAN | 0.1 | 17% | 1.8 | WGAN | 0.1 | 31% | 2.2 |
|  | 0.2 | 41% | 1.9 |  | 0.2 | 25% | 2.2 |
|  | 0.4 | 11% | 2.2 |  | 0.4 | 3% | 2.4 |

### 4.2.1 TRAINING SETTING

**Binary classifier.** We use a classifier that is similar to the GAN discriminator, consisting of a Transformer encoder followed by an averaging aggregation and linear transformation to a scalar output value. Finally, a logistic sigmoid is applied to obtain a prediction for the formula's satisfiability. The classification loss is a standard cross-entropy between real labels and predictions. Similar to the GAN discriminator, we use $n_l = 4$ layers and a batch size of $bs = 1024$. Contrary to the GAN training scheme, we use the default Transformer scheme Vaswani et al. (2017) with varying learning rate and 4000 warmup steps as well as the Adam optimizer Kingma & Ba (2015) with parameters $\beta_1 = 0.9, \beta_2 = 0.98$. This training scheme resulted in a faster improvement and higher final accuracy than adopting the settings from GAN training. We trained the classifier for 30K steps.

**Generated dataset.** To obtain a dataset of generated instances, we first train a WGAN with default parameters but smaller batch size of $512$ on a set of 10K instances from the `LTLbase` dataset. After training for 15K steps, we collect 800K generated formulas from it and call this dataset `Generated-raw`. This set is processed similar to the original base dataset: Duplicates are removed and satisfiable and unsatisfiable instances are balanced to equal amounts per formula size. We randomly keep 400K instances and call the resulting dataset `Generated`.

### 4.2.2 RESULTS

We compare the performance of similar classifiers on different training sets in Table 2. The training curves can be found in Appendix C.3. The validation accuracy is computed on the `LTLbase` dataset. Training on differently-sized subsets of `LTLbase` shows that a reduced number of training samples strongly decreases performance. 10K instances lead to immense overfitting and poor accuracy. We were not able to train a classifier on this few formulas with significantly higher accuracy.

A classifier trained on the `Generated` set however achieves almost the identical validation accuracy on the base set as the classifier that was actually trained on it. Note that the GAN that created this set was trained on only 10K instances. We therefore find that the data produced by the TGAN-SR is highly valuable as it can serve as full substitute for the complete original training data even when provided with much fewer examples.

Two instances of `LTLbase` (($\Box \neg a) \wedge (\Box \bigcirc a)$ and ($\Box \Box e) \wedge (\Box \neg e)$), i.e. only $0.02\%$, reappear in the 800K large data set `Generated-raw`. Additionally, in `Generated-raw`, only 2.3K of the 800K ($0.28\%$) generated formulas were duplicates, which displays an enormous degree of variety.

### 4.3 UNCERTAINTY OBJECTIVE FOR GENERATING HARDER-TO-CLASSIFY INSTANCES

In this experiment, we show that, by adding an uncertainty measure to a simultaneously trained classifier, the model generates instances of temporal specifications in LTL that are harder to classify. We train a model on the `LTLbase` dataset to jointly learn to imitate its formulas and classify them as satisfiable or unsatisfiable.

### 4.3.1 TRAINING SETTING

**GAN with included classifier.** For this experiment, we combine both critic and LTL satisfiability classifier into one Transformer encoder with two outputs and train them simultaneously. Both parts

Table 2: Accuracies of Transformer classifiers trained on different datasets (5-run average with standard deviations in parentheses); all are validated on the `LTLbase` dataset.

| trained on | bs | train acc @ 30K | val acc @ 30K | train acc @ 50K | val acc @ 50K |
|---|---|---|---|---|---|
| LTLbase | 1024 | 96.6% (0.5) | 95.5% (0.4) | 98.1% (0.3) | **96.1%** (0.3) |
|  | 512 | 92.4% (0.7) | 93.0% (0.8) | 95.4% (0.5) | 95.0% (0.8) |
| LTLbase 100K | 512 | 95.3% (0.7) | 88.3% (0.9) | 98.1% (0.3) | 87.8% (1.0) |
| LTLbase 10K | 512 | 100% (0.1) | 76.4% (1.7) | 100% (0.0) | 75.5% (1.5) |
| Generated | 1024 | 95.4% (0.2) | 93.6% (1.0) | 97.1% (0.1) | **93.9%** (0.3) |

share the three lower layers of the encoder but have separate fourth layers. We found this to improve both classification accuracy and GAN performance slightly compared to sharing all layers (the linear projection layer is never shared). A comparision can be found in Section C.1 in the appendix. We stick to the WGAN training scheme from Section 4.1.1 including the optimizer settings, but add an additional classification loss term similar to Section 4.2.1. The classification loss is added to the GAN critic loss and scaled by coefficient $\alpha_{\text{class}}$, which we set to 10 when training a WGAN. The resulting model achieves similar generative performance to the pure WGAN but is limited to a classification accuracy of around 92%.

**Classifier uncertainty.** We calculate the entropy of a class prediction $s$ of the classifier as $H(s) = -s \cdot \log(s) - (1 - s) \cdot \log(1 - s)$, as a measure of uncertainty on a particular instance. We add a term $-\alpha_{\text{unct}}H(s)$ to the generator's loss function, which leads to the uncertainty measure being propagated back through the critic just like the standard GAN objective. $H(s)$ is maximized at $s = 0.5$ (with value $\log 2$), so the generator is encouraged to produce instances which "confuse" the classifier included in the critic. Naturally, this conflicts with the original GAN objective, so they must be carefully balanced. As default, we chose $\alpha_{\text{conf}} = 2$. Since GAN training is hindered by adding the uncertainty objective, we only apply it after pre-training for 30K steps with default WGAN and classification objectives. We then train for additional 15K steps with the uncertainty objective included. This decreases the fraction of fully correct formulas to around 10%; sequence entropy as classification accuracy remain unaffected. From the fully trained model, we obtain a dataset similar to Section 4.2.1 and call it `Uncert-e`. Additionally, we construct a dataset of 200K formulas from this set and 200K from `LTLbase` and call it `Mixed-e`.

**Alternative uncertainty objective.** The entropy becomes unhandy to compute for values close to 0 and 1. We therefore explore a pragmatic alternative measure for (un)certainty: the absolute value of the classification logit. Values close to zero lead to predictions around $0.5$. We therefore add a generator loss of the form $\alpha_{\text{unct}}|l|$ for this variant (with $l$ the classification logit; $s = \sigma(l)$) and use a value of $\alpha_{\text{conf}} = 0.5$ in this case. The model is trained similar to the entropy variant and behaves very similarly. We call the dataset obtained from this model `Uncert-a` and also construct a mixed set `Mixed-a`.

### 4.3.2 Results

We compare the accuracy of classifiers trained similar to Section 4.2 (pure classifiers with optimized training schedule, *not* included GAN classifiers) on different (generated) datasets in Table 3. The classifier trained on `LTLbase` serves as reference again with 94.5% accuracy. Training on the `Uncert` sets however allows the classifier to achieve only 91% and 90.5% accuracy (for entropy and absolute variants, respectively). Also when trained longer than 30K steps, there is no significant improvement.

The datasets produced by WGANs with added uncertainty objective are indeed harder to classify than the original dataset `LTLbase`. To validate this, we also trained classifiers on `Mixed` sets and find that they also achieve 4.5 percent points higher accuracy when tested on the base set compared to the generated sets. Additionally, the performance on the original dataset is never deteriorated and even slightly higher when training on the mixed set. This approach is especially useful in the domain of symbolic reasoning, because data can, in contrast to archetypal deep learning domains, often be labeled automatically (e.g. with classical tools and algorithms). This underpins the usefulness of a GAN setting when generating new training instances for symbolic reasoning problems.

Table 3: Performance of classifiers trained and tested on datasets generated with uncertainty objectives; 30K steps, 5-run average with standard deviations, *not* smoothed.

| trained on | tested on | accuracy | trained on | tested on | accuracy |
|---|---|---|---|---|---|
| LTLbase | LTLbase | 94.8% (0.3) | | | |
| Uncert-e | Uncert-e | 91.0% (0.5) | Uncert-a | Uncert-a | 90.5% (0.5) |
| Mixed-e | Uncert-e | **90.2%** (0.9) | Mixed-a | Uncert-a | 89.6% (0.4) |
| Mixed-e | LTLbase | **95.3%** (0.4) | Mixed-a | LTLbase | 94.1% (0.4) |

## 5 RELATED WORK

**GANs.** Generative Adversarial Networks have been applied to discrete domains especially for text generation in a reinforcement learning setting (Chen et al., 2018; Yu et al., 2017; Che et al., 2017; Lin et al., 2017; Fedus et al., 2018; Guo et al., 2018) or by using a Gumbel softmax (Kusner & Hernández-Lobato, 2016; Zhang et al.). Kumar & Tsvetkov (2020) use a continuous, pre-trained embedding. Gulrajani et al. (2017) showed that it is possible to directly use a soft one-hot representation without any sampling. Close related work is Huang et al. (2020) and Zeng et al. (2020) for adversarial text generation. They also combine Transformers and in an adversarial learning setting, where the former rely on Gumbel softmax tricks and the latter extract a style code from reference examples. Transformers and GANs have also been combined in the domain of computer vision (Vondrick & Torralba, 2017; Jiang et al., 2021; Hudson & Zitnick, 2021). GANs have been used for data augmentation, especially for images, e.g., (Antoniou et al., 2018; Bowles et al., 2018).

**Temporal logics.** Temporal logics have been studied in computer science since their introduction by Pnueli (1977). Since then, many extensions have been developed: e.g., computation tree logic CTL and CTL$^*$ (Clarke & Emerson, 1981; Emerson & Halpern, 1986), signal temporal logic STL (Maler & Nickovic, 2004), or temporal logics for hyperproperties, e.g., HyperLTL, (Clarkson et al., 2014). Verification methods for temporal logics have been studied extensively over the years, e.g., LTL satisfiability (Li et al., 2013; Rozier & Vardi, 2007; Schuppan & Darmawan, 2011; Li et al., 2013; 2014; Schwendimann, 1998), LTL synthesis (Finkbeiner & Schewe, 2005; 2013; Bohy et al., 2012; Faymonville et al., 2017; Meyer et al., 2018), model checking (Clarke et al., 1986), or monitoring (Clarke et al., 2001; Bauer et al., 2011; Finkbeiner & Sipma, 2004; Donzé et al., 2013).

**Mathematical reasoning in machine learning.** Other works have studied datasets derived from automated theorem provers (Blanchette et al., 2016; Loos et al., 2017; Gauthier et al., 2021), interactive theorem provers (Irving et al., 2016; Kaliszyk et al., 2017; Bansal et al., 2019; Huang et al., 2019; Yang & Deng, 2019; Polu & Sutskever, 2020; Wu et al., 2021b; Li et al., 2020; Lee et al., 2020; Urban & Jakubuv, 2020; Rabe et al., 2021; Paliwal et al., 2020; Rabe & Szegedy, 2021), symbolic mathematics (Lample & Charton, 2020; Zaremba et al., 2014; Allamanis et al., 2017; Arabshahi et al., 2018), and mathematical problems in natural language (Saxton et al., 2019; Schlag et al., 2019). Learning has been applied to mathematics long before the rise of deep learning. Earlier works focused on ranking premises or clauses (Cairns, 2004; Urban, 2004; 2007; Urban et al., 2008; Meng & Paulson, 2009; Schulz, 2013; Kaliszyk & Urban, 2014).

**Neural architectures for logical reasoning.** Wu et al. (2021a) present a reinforcement learning approach for interactive theorem proving. NeuroSAT (Selsam et al., 2019) is a graph neural network (Scarselli et al., 2009; Li et al., 2018; Gilmer et al., 2017; Wu et al., 2021c) for solving the propositional satisfiability problem. A simplified NeuroSAT architecture was trained for unsat-core predictions (Selsam & Bjørner, 2019). Neural networks have been applied to 2QBF (Lederman et al., 2020), logical entailment (Evans et al., 2018), SMT (Balunovic et al., 2018), and temporal logics (Hahn et al., 2021; Schmitt et al., 2021).

## 6 CONCLUSION

We studied the capabilities of (Wasserstein) GANs equipped with two Transformer encoders to generate sensible training data for symbolic reasoning problems. We showed that both can be trained directly on the one-hot encoding space when adding Gaussian noise. We exemplary conducted experiments in the domain of symbolic mathematics and hardware specifications in temporal logics. We showed that training data can indeed be generated and that the data can be used as a meaningful substitute when training a classifier. Furthermore, we showed that a GAN setting has a speciality: by adding an uncertainty measure to the generator's output, the models generated instances on which a classifier was harder to train on. In general, logical and mathematical reasoning with neural networks requires large amounts of sensible training data. Better datasets will lead to powerful neural heuristics and end-to-end approaches for many symbolic application domains, such as mathematics, search, verification, synthesis and computer-aided design. This novel, neural perspective on the generation of symbolic reasoning instances is also of interest to generate data for tool competitions, such as SAT, SMT, or model checking competitions.

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

## A    Syntax and Semantics of Linear-time Temporal Logic (LTL)

In this section, we provide the formal syntax and semantics of Linear-time Temporal Logic (LTL). The formal syntax of LTL is given by the following grammar:

$$\varphi ::= p \mid \neg\varphi \mid \varphi \wedge \varphi \mid \bigcirc\varphi \mid \varphi \,\mathcal{U}\, \varphi,$$

where $p \in AP$ is an atomic proposition. Let $AP$ be a set of *atomic propositions*. A (*explicit*) *trace* $t$ is an infinite sequence over subsets of the atomic propositions. We define the set of traces $TR := (2^{AP})^{\omega}$. We use the following notation to manipulate traces: Let $t \in TR$ be a trace and $i \in \mathbb{N}$ be a natural number. With $t[i]$ we denote the set of propositions at $i$-th position of $t$. Therefore, $t[0]$ represents the starting element of the trace. Let $j \in \mathbb{N}$ and $j \geq i$. Then $t[i, j]$ denotes the sequence $t[i]\,t[i+1]\ldots t[j-1]\,t[j]$ and $t[i, \infty]$ denotes the infinite suffix of $t$ starting at position $i$.

Let $p \in AP$ and $t \in TR$. The semantics of an LTL formula is defined as the smallest relation $\models$ that satisfies the following conditions:

$$
\begin{array}{lll}
t \models p & \text{iff} & p \in t[0] \\
t \models \neg\varphi & \text{iff} & t \not\models \varphi \\
t \models \varphi_1 \wedge \varphi_2 & \text{iff} & t \models \varphi_1 \text{ and } t \models \varphi_2 \\
t \models \bigcirc\varphi & \text{iff} & t[1, \infty] \models \varphi \\
t \models \varphi_1 \,\mathcal{U}\, \varphi_2 & \text{iff} & \text{there exists } i \geq 0 : t[i, \infty] \models \varphi_2 \\
& & \text{and for all } 0 \leq j < i \text{ we have } t[j, \infty] \models \varphi_1
\end{array}
$$

There are several derived operators, such as $\Diamond\varphi \equiv true\,\mathcal{U}\,\varphi$ and $\square\varphi \equiv \neg\Diamond\neg\varphi$. $\Diamond\varphi$ states that $\varphi$ will *eventually* hold in the future and $\square\varphi$ states that $\varphi$ holds *globally*. Operators can be nested: $\square\Diamond\varphi$, for example, states that $\varphi$ has to occur infinitely often.

In contrast to propositional logic (SAT), where a solution is a variable assignment, the solution to the satisfiability problem of an LTL formula is a computation trace. Traces are finitely represented in the form of a "lasso" $uv^{\omega}$, where $u$, called prefix, and $v$, called period, are finite sequences of propositional formulas. For example the mutual exclusion formula above is satisfied by a trace $(\{access_{p0}\}\{access_{p1}\})^{\omega}$ that alternates indefinitely between granting process 0 ($p0$) and process 1 ($p1$) access. There are, however, infinite solutions to an LTL formula. The empty trace $\{\}^{\omega}$, where no access is granted at all, is also a solution. In our data representation, both, the LTL formula and the solution trace are represented as a finite sequence.

## B    Data Generation Details

### B.1    Rich LTL Pattern Concatenation

Previously, LTL formula generation based on patterns worked by concatenating random instantiations of a fixed set of typical specification patterns (Hahn et al., 2021). The instantiations were single variables, i.e. the response pattern $S \rightarrow \Diamond T$ could be used like $d \rightarrow \Diamond a$. We keep the concept of concatenating such patterns, but extend the process by mainly two concepts: rich pattern instantiations and groundings.

Dwyer et al. (1999) analyzed typical specifications constructed a system of frequently occurring patterns. They are grouped into different types such as *absence* ($\neg S$, something does not occur) or response ($S \rightarrow \Diamond T$, if $S$ occurred, $T$ must eventually respond). These patterns can again appear in different scopes such as globally, before or between some events. The global absence pattern is then $\square\neg S$; the absence before $Q$ pattern reads $Q \,\mathcal{R}\, \neg S$. When generating a new pattern for concatenation, we sample both a type and a scope and assign different probabilities to account for more common and exotic combinations. Additionally, we instantiate patterns not with single variables, but full subformulas, which results in much more reasonable and interesting patterns such as $\square\neg(a \wedge b)$ or $((\neg d \vee \bigcirc b) \rightarrow \Diamond(c \wedge f)) \,\mathcal{U}\, e$. These subformulas may still contain temporal operators, but are strongly biased towards pure boolean operators.

During concatenating the different parts of a formula, we also distinguish between adding instantiated patterns and *groundings*. The problem with complex patterns and especially complex scopes

is that they must be "activated" to have some effect: If some constraint must only hold between $Q$ and $R$ but these events never happen, the whole pattern is effectively useless. A grounding is a term that is likely to activate scopes, such as $\bigcirc\bigcirc a \wedge \neg b$ or $\square\diamondsuit c$. The variables used here are also biased to coincide with the ones already used in previous patterns to further increase the change for dependencies. Groundings are added with 45% probability instead of a specification pattern.

We observe that these changes indeed lead to a much higher chance of unsatisfiability. Consider the code in `data_generation/spec_patterns.py` for exact reference of the individual steps in the generation process.

### B.2  Temporal Relaxation for Formula Inspection

We inspect the unsatisfiable formulas obtained by our generation process more closely. Concretely, we want to make sure that unsatisfiabilities do not stem from simple boolean contradictions, but actually require temporal reasoning to some extend. For example, the formula $(a \vee b) \wedge \neg b \wedge \square \neg a$ can be found to be unsatisfiable without considering multiple time steps. In contrast, this would be required for a formula like $\neg a \,\mathcal{U}\, b \wedge \square \neg b \wedge \diamondsuit a$.

We therefore introduce a *temporal relaxation* that transforms a LTL formula into a purely boolean formula. This allows us to check whether the relaxed version is already unsatisfiable (so, no temporal reasoning is required) or if it is only temporally unsatisfiable, which is the desired outcome. The relaxation is defined as follows:

$$
\begin{aligned}
Rel(\varphi * \psi) &= Rel(\varphi) * Rel(\psi) \quad \text{for } * \in \{\wedge, \vee, \rightarrow, \leftrightarrow, \oplus\} \\
Rel(\varphi * \psi) &= Rel(\varphi) \vee Rel(\psi) \quad \text{for } * \in \{\mathcal{U}, \mathcal{W}\} \\
Rel(\varphi \,\mathcal{R}\, \psi) &= Rel(\psi) \\
Rel(\bigcirc \varphi) &= \top \\
Rel(\square \varphi) &= \varphi \\
Rel(\diamondsuit \varphi) &= \top \\
Rel(\alpha) &= \alpha \quad \text{for } \alpha \in AP \cup \{\top, \bot\} \\
Rel(\neg\alpha) &= \neg\alpha \quad \text{for } \alpha \in AP \cup \{\top, \bot\}
\end{aligned}
\tag{1}
$$

Notably, negation is only allowed at the level of atoms. Each LTL formula can be rewritten in a negation normal form (NNF), where only operators $\wedge, \vee, \mathcal{U}, \mathcal{R}, \bigcirc$ occur anywhere and negations only before atoms. Consequently, the relaxation can be applied to each LTL formula by first bringing it to NNF.

### B.3  Base Dataset

We generated a raw dataset of 1.6M instances (see reproducibility section for details) up to size 50. To determine satisfiability, we use the tool `aalta` (Li et al., 2014). Its length distribution and satisfiability distribution is shown in Figures 5 and 6.

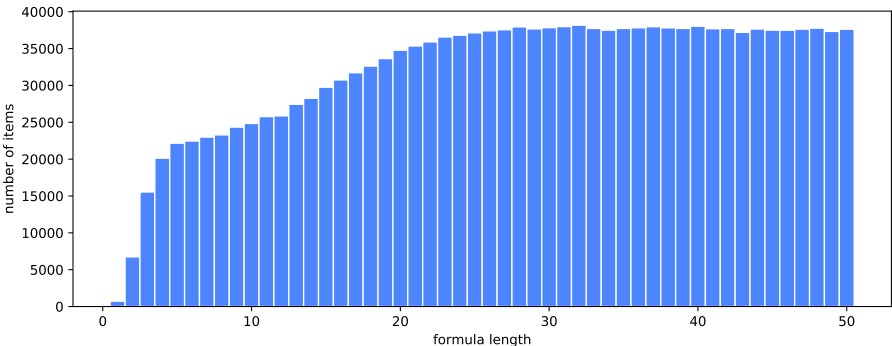

Figure 5: Raw dataset size distribution

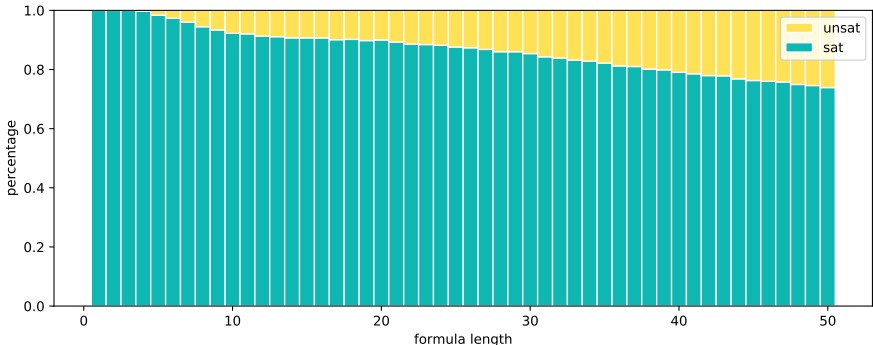

Figure 6: Raw dataset satisfiability proportions

We filter out duplicates and balance satisfiable and unsatisfiable instances per size (Figure 7). Additionally, we apply the temporal relaxation and determine the satisfiability of relaxed unsatisfiable instances. This distinction is included in Figure 8. Finally, the dataset is split into a training set (80%) and validation set (10%). The resulting training set contains around 380K instances.

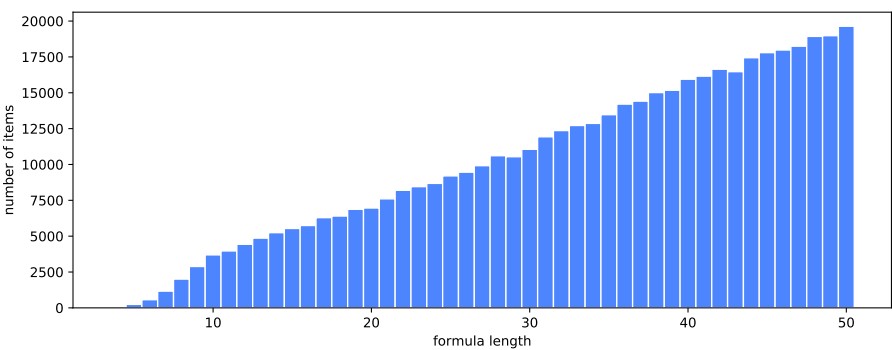

Figure 7: Final dataset size distribution (average size 34.6)

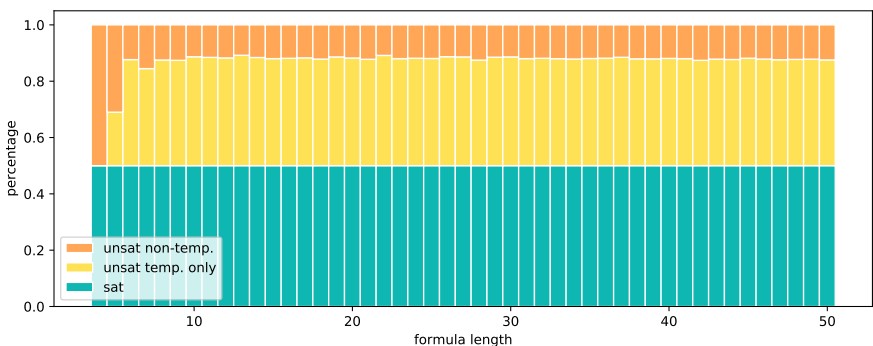

Figure 8: Final dataset satisfiability proportions

## B.4 WGAN-GENERATED DATASETS

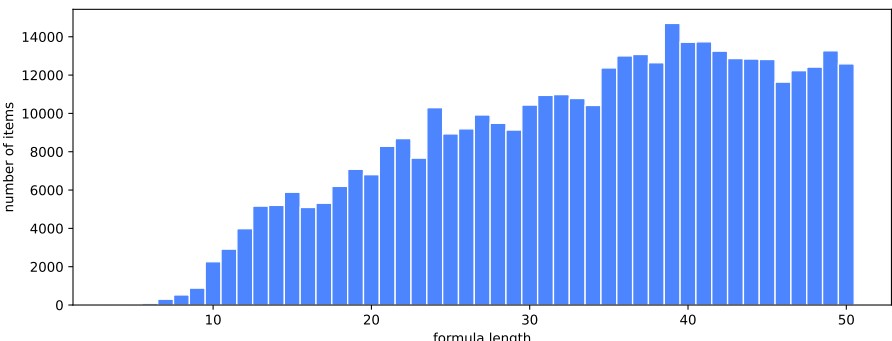

Figure 9: `Generated` dataset size distribution (average size 33.6)

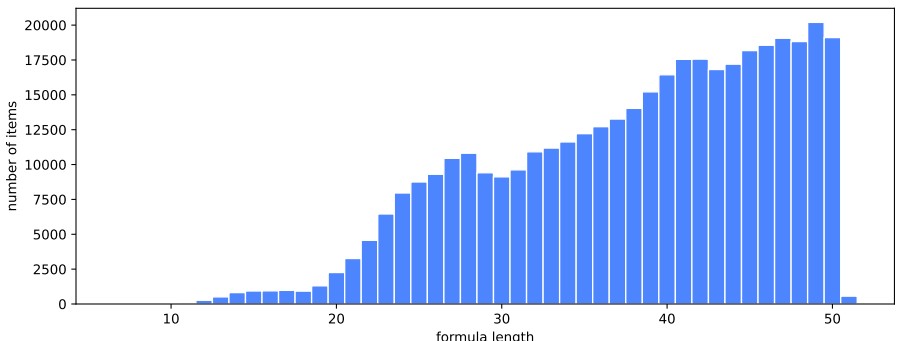

Figure 10: `Uncert-e` dataset size distribution (average size 38.0)

## C ADDITIONAL EXPERIMENTS AND INFORMATION

### C.1 SHARED LAYERS FOR CLASSIFIER INCLUDED IN CRITIC

Table 4: Different number of shared layers for WGAN with included classifier, 2 runs each, 30K steps

| shared layers | se | fc | val acc |
|---|---|---|---|
| 0 / 4 | 2.2 | 31.8% (0.2) | 89.9% (2.3) |
| 2 / 4 | 2.2 | 26.6% (1.7) | 92.5% (0.1) |
| 3 / 4 | 2.2 | 24.9% (0.3) | 92.1% (0.6) |
| 4 / 4 | 2.2 | 24.0% (1.2) | 90.5% (0.5) |

Table 4 shows classification benefits for sharing only some layers between classifier and critic. Also note that not sharing any layers, while yielding the highest fraction of fully correct formulas in the joint GAN and classification objective, degrades performance in the uncertainty setting, where a loss is backpropagated through the classifier part.

### C.2 HYPER-PARAMETER COMPARISON

A hyper-parameter comparison with a 2-run average at 15K training steps.

| $n_{lG}$ | $n_{lD}$ | $n_c$ | $bs$ | variant | fully correct |
|---|---|---|---|---|---|
| 2 | 2 | 1 | 1024 | GAN | 6% |
| 2 | 2 | 1 | 1024 | WGAN | 4% |
| 2 | 2 | 2 | 512 | GAN | 6% |
| 2 | 2 | 2 | 512 | WGAN | 4% |
| 2 | 2 | 2 | 1024 | GAN | 7% |
| 2 | 2 | 2 | 1024 | WGAN | 5% |
| 2 | 2 | 2 | 2048 | GAN | 5% |
| 2 | 2 | 2 | 2048 | WGAN | 6% |
| 2 | 2 | 3 | 1024 | WGAN | 5% |
| 2 | 4 | 2 | 1024 | GAN | 8% |
| 2 | 4 | 2 | 1024 | WGAN | 9% |
| 3 | 3 | 2 | 1024 | GAN | 8% |
| 3 | 3 | 2 | 1024 | WGAN | 13% |
| 4 | 2 | 2 | 1024 | GAN | 7% |
| 4 | 2 | 2 | 1024 | WGAN | 14% |
| 4 | 4 | 2 | 1024 | GAN | 10% |
| 4 | 4 | 2 | 1024 | WGAN | 16% |
| 6 | 4 | 2 | 1024 | GAN | 17% |
| 6 | 4 | 2 | 1024 | WGAN | 20% |
| 6 | 6 | 2 | 1024 | WGAN | 15% |
| 8 | 6 | 2 | 1024 | WGAN | 18% |

## C.3 TRAINING CURVES FOR DATA SUBSTITUTION EXPERIMENTS

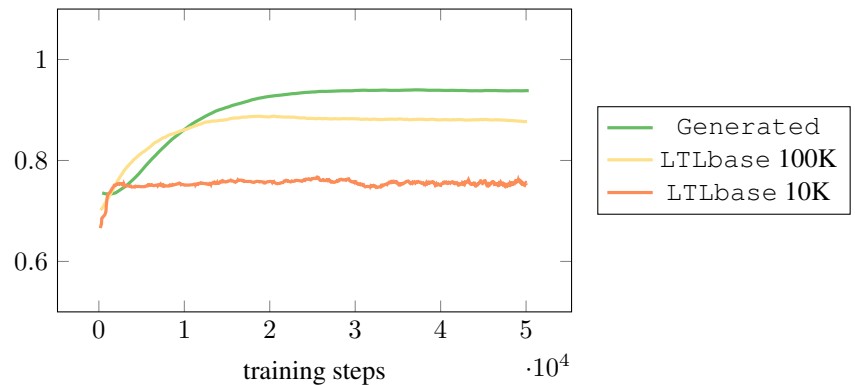

Figure 11: Validation accuracy during training of Transformer classifiers on different datasets. 5-run average, smoothed ($\alpha = 0.9$). Complements Table 2.

We provide the training curves for the data substition experiment (see Figure 11).

## C.4 STANDARD DEVIATIONS FOR TABLE 1

Table 5: Standard deviations for Table 1, 3-run average, smoothed ($\alpha = 0.95$)

| architecture | $\sigma_{real}$ | $fc$ sd | $se$ sd | | architecture | $\sigma_{real}$ | $fc$ sd | $se$ sd |
|---|---|---|---|---|---|---|---|---|
|  | 0.00 | 0.00 | - | |  | 0 | 1.51 | 0.00 |
|  | 0.05 | 4.10 | 0.04 | |  | 0.05 | 2.52 | 0.00 |
| GAN | 0.1 | 6.50 | 0.03 | | WGAN | 0.1 | 1.08 | 0.01 |
|  | 0.2 | 3.92 | 0.04 | |  | 0.2 | 0.47 | 0.00 |
|  | 0.4 | 0.14 | 0.02 | |  | 0.4 | 0.14 | 0.03 |

We provide the standard deviations for Table 1 across 3 runs (see Table 5).

## C.5 GAN WITH UNIFORM NOISE

Table 6: GAN variant with uniform instead of Gaussian noise. 2-run average with standard deviations, smoothed ($\alpha = 0.99$)

| min | max | $fc$ | $se$ |
|---|---|---|---|
| 0 | 0.1 | 19.2% (2.94) | 1.6 (0.19) |
| 0 | 0.2 | 33.3% (1.04) | 1.8 (0.01) |
| 0 | 0.4 | 11.2% (3.32) | 2.0 (0.07) |

As evident from Table 6 in comparison with Table 1, a uniform noise has no benefit over Gaussian noise.

## C.6 OUT-OF-DISTRIBUTION CLASSIFICATION EXPERIMENTS

Table 7: Classifiers trained on different datasets tested out-of-distribution. 5-run average, not smoothed

| trained on | training steps | tested on | accuracy |
|---|---|---|---|
| LTLbase | 30K | Benchmarks | 85.2% (2.2) |
| Uncert-e | 30K | Benchmarks | 85.9% (2.9) |
| Uncert-e | 30K | LTLbase | 87.5% (0.9) |
| Mixed-e | 30K | Mixed-e | 92.7% (0.6) |
| LTLbase | 50K | Benchmarks | 86.0% (5.0) |
| Generated | 50K | Benchmarks | 94.1% (1.2) |

A synthetic dataset that is designed to bring classical solvers to their limits is a portfolio dataset (Schuppan & Darmawan, 2011), of which around 750 formulas fit into our encoder token and size restrictions. We conducted an out-of-distribution test on these scalable benchmarks (Table 7). Note that almost all of the instances are satisfiable.

