# OpenReview forum: "Generating Symbolic Reasoning Problems with Transformer GANs"
_ICLR.cc/2022/Conference — ICLR 2022 Submitted_

### Official Review · Reviewer_JuGn · 2021-10-25

**Correctness:** 4
**Technical Novelty And Significance:** 3
**Empirical Novelty And Significance:** 3
**Recommendation:** 8
**Confidence:** 4

**Main Review:**

This is a strong paper overall: innovative, well-written, and potentially important.

- I found it awkward that the prose stressed that two domains would be considered, in some cases even listing symbolic mathematics (i.e. function integration) first, even though the integration domain received much shorter shrift and most of the experiments seem to be LTL only. I think the paper would be stronger with two domains throughout rather than one.

- How many examples were in LTLBase? It might be nice to see an estimate of the effective size of the "Generated" dataset with respect to the original distribution.

- I see that in S4.2 you are starting with only 10k samples from LTLBase, but what are you training on exactly for S4.1? If a larger dataset than in S4.2, did you do a similar check for duplicates between the generated examples and the training dataset? How many (if any) of the 30% that are syntactically valid were seen during training?

- It is not clear to me whether the last paragraph of S4.1 is describing a different training regime than the paragraph preceeding it.

- The first sentence seems more suitable for a blog post than a conference paper; in particular, the claim that deep learning is "on the verge" of something sounds unscientific. Also, the word "transitioning" may not be appropriate, since presumably deep learning will continue to be applied to e.g. image recognition.

- I particularly liked the approach in S4.3 in which an additional objective is introduced that shifts the generated distribution away from the original one. In general, the true objective of data augmentation may be broader than simply modeling the specific distribution of training examples one has at hand. Have the authors considered other "knobs" to add that would allow generating more diverse examples, that may provide useful augmentation either for the original distribution or for out-of-distribution evaluations?

- Minor comment: I did not find that it added much to the paper to consider both GANs and WGANs. I would suggest only discussing WGANs.


**Summary Of The Paper:**

The authors apply (W)GANs with transformer encoders for data-augmentation in two symbolic domains: LTL and function integration. There are many interesting findings, three of which stand out. First, in both domains, the network learns to generate syntactically-correct examples roughly 30% of the time. Second, for LTL, a GAN trained on a dataset of size 10k can produce a much larger dataset, such that training on the new dataset is almost as good as training on the original distribution (when evaluating on the original distribution). Third, by also rewarding the generator for confusing a classifier, they can generate new problems that are harder to classify than those in the original distribution.

**Summary Of The Review:**

This is a strong paper overall: innovative, well-written, and potentially important. I think it will be of interest to anyone applying machine learning in data-sparse symbolic domains.

---

> ### Author Response · Authors · 2021-11-12
> **Reply to Reviewer JuGn**
>
> We thank the reviewer for their thorough review and valuable feedback to improve the submission. We will address the questions in the following.
>
> > I found it awkward that the prose stressed that two domains would be considered, in some cases even listing symbolic mathematics (i.e. function integration) first, even though the integration domain received much shorter shrift and most of the experiments seem to be LTL only. I think the paper would be stronger with two domains throughout rather than one.
>
> Symbolic integration is not a classification, but a complex translation task. We chose the integration dataset to validate our main GAN findings for generating syntactically consistent formulas on an existing dataset. We will clarify this in the paper.
>
> &nbsp;
>
> > How many examples were in LTLBase? It might be nice to see an estimate of the effective size of the "Generated" dataset with respect to the original distribution.
>
> The training set of LTLbase is slightly larger than 380K instances.
>
> &nbsp;
>
> > I see that in S4.2 you are starting with only 10k samples from LTLBase, but what are you training on exactly for S4.1? If a larger dataset than in S4.2, did you do a similar check for duplicates between the generated examples and the training dataset? How many (if any) of the 30% that are syntactically valid were seen during training?
>
> In Sections 4.1 and 4.3 we use the full LTLbase set. We will clarify this in the paper.
> We will also calculate the duplicates and copies of existing instances for this set and add it to the paper. We found this to be particularly interesting for the very small data set from which we expected more duplicates.
>
> &nbsp;
>
> > It is not clear to me whether the last paragraph of S4.1 is describing a different training regime than the paragraph preceeding it.
>
> If the reviewer is referring to the paragraph titled “effects of additive noise on one-hot representation”: This is the exact same training regime, we simply additionally computed the argmaxed versions and plugged them into the discriminator (without training on those). This could be easily misunderstood, thanks for pointing this out. We will clarify this.
>
> &nbsp;
>
> > The first sentence seems more suitable for a blog post than a conference paper; in particular, the claim that deep learning is "on the verge" of something sounds unscientific. Also, the word "transitioning" may not be appropriate, since presumably deep learning will continue to be applied to e.g. image recognition.
>
> Thanks for the feedback, we will rephrase it.
>
> &nbsp;
>
> > I particularly liked the approach in S4.3 in which an additional objective is introduced that shifts the generated distribution away from the original one. In general, the true objective of data augmentation may be broader than simply modeling the specific distribution of training examples one has at hand. Have the authors considered other "knobs" to add that would allow generating more diverse examples, that may provide useful augmentation either for the original distribution or for out-of-distribution evaluations?
>
> We plan to expand the approach by continuously adding newly generated and labeled instances to the original dataset during training so that the target distribution moves permanently.

---

> > ### Comment · Reviewer_JuGn · 2021-11-29
> > **Re: author response**
> >
> > I thank the authors for their response and for providing additional information about the experiments. I still don't find generating syntactically-correct formulae alone to be particularly interesting, since it is easy to randomly generate well-formed formulae. I would still like to see that the generated data has statistical value (as was shown for LTL).

---

### Official Review · Reviewer_qNn9 · 2021-10-31

**Correctness:** 3
**Technical Novelty And Significance:** 2
**Empirical Novelty And Significance:** 3
**Recommendation:** 3
**Confidence:** 4

**Main Review:**

### Strengths

* Simple idea: using GANs for data augmentation.

* Empirically proving that GANs can generate parsable instances and that GANs
  can generate interesting problems that are hard to solve and contribute to
  improve the performance of classifiers.

### Weaknesses

I have several concerns and questions as follows.

* I'm not entirely sure of how the generated data are labeled.  Is it done by
  the GAN discriminator or by another algorithm as labeling LTL formulas in the
  real dataset with the tool aalta (Appendix B.3)?

  If the latter is employed, it indicates the existence of an algorithm to
  address the task of interest, so I'm wondering why machine learning is applied
  to it.  Perhaps ML models might be expected to answer the problems faster
  than the algorithm, but labeling with the algorithm would be successful only
  on problems that the algorithm can answer in a given time; thus, it seems hard
  to label problems that the algorithm cannot solve in a reasonable time,
  although the ML models should approach such problems.

  For the former, the generated data could be mislabeled.  Then, it is not clear
  for me why training on such data can produce a model with the performance very
  close to the model trained on data with correct labels (Table 2).

* I don't think this is the first work to use GANs for data augmentation.  For
  example, [1] and [2] used GANs for augmenting image data.  Unlike
  these previous works, the paper addresses sequential data that are textual
  representations of mathematical expressions, but lacking the discussion and
  (qualitative) comparison with them makes a challenge and novelty of the paper
  unclear.

  [1] Antreas Antoniou, Amos J. Storkey, Harrison Edwards.
  Data Augmentation Generative Adversarial Networks.

  [2] Christopher Bowles, Liang Chen, Ricardo Guerrero, Paul Bentley, Roger N. Gunn, Alexander Hammers, David Alexander Dickie, Maria del C. Valdés Hernández, Joanna M. Wardlaw, Daniel Rueckert:
  GAN Augmentation: Augmenting Training Data using Generative Adversarial Networks.


* The abstract and introduction seem oversold somewhat in that they don't
  mention that the experiment conducted for symbolic mathematics only
  confirms the GAN model can produce syntactically correct problems; thus, the
  full usefulness of the GAN-based data augmentation is still open in the
  symbolic mathematics domain.  I think it would be nicer to conduct more
  experiments on symbolic mathematics, which would demonstrate the proposed
  approach is useful in broader domains.

* The experiments on satisfiability of LTL formulas look interesting, but I'm
  concerned that it is adequate to be tacked with machine learning.  As written
  in Appendix B.3, there exists a tool for checking satisfiability of LTL
  formulas, and it is used to label the (real) dataset. Then, why do we need the
  trained model for checking satisfiability?  A critical problem with the
  trained model is that its prediction might be wrong.  How can we use a
  satisfiability checker that may produce wrong answers?

  For a similar reason, I'm not sure why machine learning is useful for the
  domain where "data can ... often be labeled automatically" (page 8).

* The paper uses GAN and WGAN, but it does not investigate why they make a
  difference in the experiments (e.g., Table 1).


### Minor comments / questions

P2 "\neg \box (access_p0 \wedge access_p1)"  I think \box (globally) should be
replaced with \diamond (eventually) for correction.

P2 "Mathematical expressions"  What are these?  Are they different from random functions?

P3 "continuous domains were"  where

P4 "we use the alternative generator loss"  Why?

P6 "the origin training data"  original

P6, Table 1:
  For GAN, does the use of \sigmoid_real larger than 0.2 make the fraction of fc larger?

P7 "we combine both critic and classifier into one Transformer encoder"  Why are not they separated?

P8: The result in Table 3 seems peculiar to me.  Why does the model trained on
Mixed-e outperform the model trained on LTLbase even when tested on LTLbase?

### Post-Rebuttal

I would like to thank the authors for the additional comments to answer my questions.
However, I still have two major concerns that make me hesitate to accept the paper.
The first s that I don't still find the task of generating symbolic expressions interesting.
The second is about an application of the approach.  The response from the authors says that neural models may compute solutions faster than classical tools.  I don't disagree with this claim, but the paper doesn't show that the proposed approach is indeed helpful for that task.  I would like to see more discussions and evidence for the story to hold true (e.g., how the augmented data are labeled, whether they can improve the performance of neural models that compute solutions, etc.)

**Summary Of The Paper:**

This paper makes the use of GANs equipped with Transformers to generate new
instances of the problems in the symbolic reasoning domains, and demonstrated
the usefulness of the idea on two domains: satisfiability prediction of LTL
formulas and mathematical reasoning on integration and ordinary differential
equations.  The experiments show that the trained GAN models can produce
parsable instances on both of the domains and that training on the generated
data can improve the accuracy performance of a classification model for LTL
satisfiability prediction.

**Summary Of The Review:**

The paper addresses a critical problem in symbolic reasoning domains, and the
experimental results are promising. However, I think it's not ready for
publication because of lacking a discussion for practical settings to use the
proposed GAN-based data augmentation, an evidence to show its generality, and a
comparison with the previous work.

---

> ### Author Response · Authors · 2021-11-12
> **Reply to Reviewer qNn9**
>
> We thank the reviewer for their thorough review and valuable feedback to improve the submission. We would like to address your questions and concerns in the following. We also thank the reviewer for pointing out typos.
>
> > I'm not entirely sure of how the generated data are labeled. Is it done by the GAN discriminator or by another algorithm as labeling LTL formulas in the real dataset with the tool aalta (Appendix B.3)?
>
> The latter is the case.
>
> &nbsp;
>
> > I don't think this is the first work to use GANs for data augmentation. For example, [1] and [2] used GANs for augmenting image data. Unlike these previous works, the paper addresses sequential data that are textual representations of mathematical expressions, but lacking the discussion and (qualitative) comparison with them makes a challenge and novelty of the paper unclear.
> [...]
>
> We thank the reviewer for these pointers to related work. We will include a discussion of it in the paper.
>
> &nbsp;
>
> > The abstract and introduction seem oversold somewhat in that they don't mention that the experiment conducted for symbolic mathematics only confirms the GAN model can produce syntactically correct problems; thus, the full usefulness of the GAN-based data augmentation is still open in the symbolic mathematics domain. I think it would be nicer to conduct more experiments on symbolic mathematics, which would demonstrate the proposed approach is useful in broader domains.
>
> Symbolic integration is not a classification, but a complex translation task. We chose the integration dataset to validate our main GAN findings for generating syntactically consistent formulas on an existing dataset.
>
> &nbsp;
>
> > The experiments on satisfiability of LTL formulas look interesting, but I'm concerned that it is adequate to be tacked with machine learning. As written in Appendix B.3, there exists a tool for checking satisfiability of LTL formulas, and it is used to label the (real) dataset. Then, why do we need the trained model for checking satisfiability? A critical problem with the trained model is that its prediction might be wrong. How can we use a satisfiability checker that may produce wrong answers?
>  For a similar reason, I'm not sure why machine learning is useful for the domain where "data can ... often be labeled automatically" (page 8).
>
> Even when classical algorithms exist, having neural reasoning engines is still very desirable: classical tools often do not scale very well and suffer from a high computational complexity. Neural models, however, can make fast predictions. These can be checked by a classical tool in a fraction of the time that it would need to come up with a solution. Depending on the domain, we also expect it to be feasible when training only semi-supervised and have no labels whenever the classical tool times out. Note that for LTL, models can already predict solutions when classical solvers time out (Hahn et al. ‘20).
>
> &nbsp;
>
> > P2 "Mathematical expressions" What are these? Are they different from random functions?
>
> They are random mathematical expressions with one variable, so they correspond to random functions, yes. For details, please refer to the authors of the dataset (Lample and Charton, ICLR ’20).
>
> > P4 "we use the alternative generator loss" Why?
>
> Training is not successful otherwise.
>
> > P6, Table 1: For GAN, does the use of $\sigma_\text{real}$ larger than 0.2 make the fraction of fc larger?
>
> Thanks for pointing this out, we will expand the tables.
>
> > P7 "we combine both critic and classifier into one Transformer encoder" Why are not they separated?
>
> We found it reasonable to share layers in the architecture and expect this also to help with backpropagating the uncertainty loss. We will add a comparison, possibly to the appendix.
>
> > P8: The result in Table 3 seems peculiar to me. Why does the model trained on Mixed-e outperform the model trained on LTLbase even when tested on LTLbase?
>
> We consider this an interesting result. The generated training data seems to act as a regularizer, which is very desirable. We will also add the standard deviations between multiple runs to the paper.

---

> > ### Comment · Reviewer_qNn9 · 2021-11-24
> > **Follow-up Questions**
> >
> > Thank you for the response.  It addresses my questions in the review, but I still have several concerns.
> >
> > >> The abstract and introduction seem oversold somewhat in that they don't mention that the experiment conducted for symbolic mathematics only confirms the GAN model can produce syntactically correct problems; thus, the full usefulness of the GAN-based data augmentation is still open in the symbolic mathematics domain. I think it would be nicer to conduct more experiments on symbolic mathematics, which would demonstrate the proposed approach is useful in broader domains.
> > >
> > > Symbolic integration is not a classification, but a complex translation task. We chose the integration dataset to validate our main GAN findings for generating syntactically consistent formulas on an existing dataset.
> >
> > I'm not excited about just generating syntactically correct problems because a similar task has been approached by symbolic reasoning (as in program synthesis) and even by neural methods (e.g., as the following [1]).  Perhaps I might miss something, but I can't find from the paper nor response why the experimental result on symbolic integration is exciting or worth reporting.
> >
> > [1] Baptiste Rozière, Marie-Anne Lachaux, Lowik Chanussot, Guillaume Lample. Unsupervised Translation of Programming Languages. NeurIPS 2020.
> >
> > > Even when classical algorithms exist, having neural reasoning engines is still very desirable: classical tools often do not scale very well and suffer from a high computational complexity. Neural models, however, can make fast predictions. These can be checked by a classical tool in a fraction of the time that it would need to come up with a solution.
> >
> > Do the authors mean  checking predictions (without evidence) is (much) faster than computing solutions?  Then, I'm not convinced with this claim.  Can the authors provide references to prove it?  How broadly is the claim applicable?  If the claim is true, then why don't practitioners just run two processes of the checker by supposing the answer of a problem is satisfiable in one process and unsatisfiable in the other?  If this worked well, it would be quite surprising for me that we get the correct answer with at most twice of machine resource that is necessary to run one checking process, and that we could get the  answer very quickly by running the two processes in parallel.
> >
> > >> P4 "we use the alternative generator loss" Why?
> > >
> > > Training is not successful otherwise.
> >
> > Why not successful?  How did the authors come up with the alternative loss, and why do the authors consider it can be a replacement?

---

> > > ### Author Response · Authors · 2021-11-25
> > > **Reply to Reviewer qNn9**
> > >
> > > We thank the reviewer and appreciate the opportunity to address their concerns.
> > >
> > > > I'm not excited about just generating syntactically correct problems because a similar task has been approached by symbolic reasoning (as in program synthesis) and even by neural methods (e.g., as the following [1]). Perhaps I might miss something, but I can't find from the paper nor response why the experimental result on symbolic integration is exciting or worth reporting.
> > > [1] Baptiste Rozière, Marie-Anne Lachaux, Lowik Chanussot, Guillaume Lample. Unsupervised Translation of Programming Languages. NeurIPS 2020.
> > >
> > > Thanks for the pointer to related work, but we would like to emphasize that these are completely different settings. Our work is a study and implementation of (W)GANs equipped with transformer encoders, showing that they can generate complicated symbolic instances without any auto-regression. For demonstration, we picked two of recently published works in symbolic domains (with large enough base datasets and an instance size that fit into the models). We happily invite researchers in program synthesis to apply our approach and use our implementation to generate more data.
> > >
> > > &nbsp;
> > >
> > > > Do the authors mean checking predictions (without evidence) is (much) faster than computing solutions? Then, I'm not convinced with this claim. Can the authors provide references to prove it? How broadly is the claim applicable? If the claim is true, then why don't practitioners just run two processes of the checker by supposing the answer of a problem is satisfiable in one process and unsatisfiable in the other? If this worked well, it would be quite surprising for me that we get the correct answer with at most twice of machine resource that is necessary to run one checking process, and that we could get the answer very quickly by running the two processes in parallel.
> > >
> > > We apologize if our answer has led to confusion.
> > > Our answer was referring to when a NN is trained to also return the witness (“evidence”) directly or partially. This can also be done (but is beyond the scope of this work). For example, SAT and LTL (e.g. Selsam and Bjorner ‘19, Hahn et al. ‘21), circuit synthesis (Schmitt et al. ‘21), QBF (Lederman et al. ‘18) or work on theorem proving (see related work).
> > >
> > > In this case, the claim indeed holds true: Computing a satisfying assignment in SAT is NP-complete, while checking if an assignment is a solution is in P (Cook ‘71, Levin ‘73). Computing a solution trace to an LTL formula is PSPACE-complete (Sistla and Clarke ‘85), while checking a trace is only in NC (Kuhtz and Finkbeiner ‘09). Computing a circuit to an LTL formula is 2-EXPTIME-complete (Pnueli and Rosner 1989), while checking it is only PSPACE-complete (Sistla and Clarke ‘85). Same holds true for optimization problems, SMT, theorem proving, etc.
> > > We consider it a very promising approach to have multiple witness predictions of a NN, with a beam search for example. These can then be checked cheaply, which will result in a performance boost. Only in the worst case, when no witness prediction of the NN is a solution, one has to fall back to a classic search algorithm.
> > >
> > > &nbsp;
> > >
> > > > Why not successful? How did the authors come up with the alternative loss, and why do the authors consider it can be a replacement?
> > >
> > > The original GAN paper (Goodfellow et al. ‘14) introduces both variants. The first proposed loss is more theoretically grounded, while the alternative one often works better in practice (due to circumventing close-to-zero gradients).

---

### Official Review · Reviewer_3zMF · 2021-11-01

**Correctness:** 2
**Technical Novelty And Significance:** 2
**Empirical Novelty And Significance:** Not applicable
**Recommendation:** 5
**Confidence:** 4

**Main Review:**

The problem is interesting and the paper is well motivated.

Abstract: What do they authors mean by “synthetically generated instances are often hard to evaluate in terms of their meaningfulness”?

What do the authors mean by “training on randomly generated data carries the risk of training on meaningless data or the risk of introducing unwanted biases”? Provided that the conclusion follow logically, can the authors elaborate on what they mean here?

“We show that training directly in the one-hot encoding space is possible when adding Gaussian noise to each position”. Why use Gaussian noise? The authors should be more clear about why they add noise to the real samples? Presumably this is to make distinguishing the real samples from the fake ones non-trivial? This is only explained later on and is a bit confusing.

“on which a classifier can successfully be trained on”. Do you test on an established dataset?

“The generator’s input is a real scalar random value with uniform distribution [0, 1] for each position.” It is not clear here what is mean by “each position”. I assume this is each symbol in the input? But this is not clearly explained.

“The position-wise padding mask is copied from the real data during training, so the lengths of real and generated formulas at the same position in a batch are always identical.” The first time you refer to **the** padding mask it is not clear what this is or where it comes from?

“Still, both generators are able to produce a large fraction of fully correct temporal specifications, which we find surprising” Could there be overfitting? Are the results the same across many runs? Is 0.3 a large fraction? How does this compare to generating examples randomly? This would make a good baseline (even if your models perform worse).

Figure 4: Interesting results.

The satisfiability classifier results are interesting. What would be better tho, is to show that training on the generated data improved results on an established dataset.

It is interesting that you can learn on data generated using the LTLbase 10k dataset and perform better on the validation set than training directly on the LRLbase (and perform similarly well when training on the whole LTLbase dataset). However, it’s clear that the model has overfit to the LTLbase 10k. How did you decide to stop training? Does this happen for all runs? It would be helpful to see the training and test curves while the model is training. Additionally, it would be good to see a graph with multiple runs.

For all results in the paper, it would be best to perform multiple runs and report the standard deviations.

In the section titled “GAN with included classifier”: What are you classifying? Are you predicting satisfiability? This is not clear. How do you know the satisfiability of the generated samples? Is there a 50/50 split of satisfiable and unsatisfiable examples?

Are there the same number of training examples in the LTLbase, Uncert-e and mixte-e datasets?

Table 3: Are you training and testing on different splits of the dataset? It would help to add the std? The results somewhat suggest that training to increase uncertainty does produce slightly more challenging problems, but it’s hard to tell because it’s not clear how much results vary between runs and it’s not clear if the differences are statistically significant.

Table 3: It’s very important to know how train and test samples were split for Uncert-e. Uncert-e being more difficult to classify could also be explained by lack of variation and the samples you tested happening to be in a different mode to those in the training set. What is the average length of problems in each data set?

Is there a qualitative difference between samples generated with and without the uncertainty loss? What makes the problems harder?

Does training on Uncert-e improve performance on LTLbase (more so than training on generated?) Is there a standard dataset that you can show improvement on?

Why are there not more quantitative results on the integration examples? It’s clear that a classifier can already perform very well on the LTL tasks, perhaps it would be easier to see improvements on the function integration task? It does not appear that there are any results for this in the main text?

General: The paper does not clearly separate LTL results and symbolic math results. This makes some results harder to parse.

General: This model does not generate a supervised training dataset since you still need to use existing algorithms/ programs to compute the labels/targets. This could be a problem for datasets where the solution is intractable and would also suggest that you already have a model capable of solving the problem for which you are generating the data. What is the long term motivation of this work if you either (a) cannot generate labels/targets or (b) can already use existing algorithm to solve these problems.


**Summary Of The Paper:**

Generating symbolic reasoning training data using transformer GANs. Use classifier uncertainty in the generator objective to generate samples that are harder for the classifier to solve than the original data. The problem is very interesting, but the empirical results could be improved.

**Summary Of The Review:**

The problem is really interesting.


Correctness:
There are problems with the experimental results. If the authors can add the suggested results and show that their results are statistically significant, I would be very happy to increase my score.

Novelty: The approach also lacks novelty, only proposing an additional loss which is not clearly described. However, their application is very interesting.

---

> ### Author Response · Authors · 2021-11-12
> **Reply to Reviewer 3zMF, part 1**
>
> We thank the reviewer for their thorough review and valuable feedback to improve the submission. We will answer your questions in the following.
>
> > What do they authors mean by “synthetically generated instances are often hard to evaluate in terms of their meaningfulness” [and] by “training on randomly generated data carries the risk of training on meaningless data or the risk of introducing unwanted biases”?
>
> We mean that completely random symbolic inputs, such as random formula trees, do not reflect the distribution of real-world problem instances. We will clarify this in the paper.
>
> &nbsp;
>
> > “We show that training directly in the one-hot encoding space is possible when adding Gaussian noise to each position”. Why use Gaussian noise? The authors should be more clear about why they add noise to the real samples? Presumably this is to make distinguishing the real samples from the fake ones non-trivial? This is only explained later on and is a bit confusing.
>
> Exactly, this is to diminish identification of real/fake samples by their representation in one-hot space alone. As outlined in the end of Section 4.1.2 and Table 1, training fails without noise in a pure GAN setting. There is no particular motivation for using Gaussian noise. We will run additional experiments with other types of noise and report the results as soon as possible.
>
> &nbsp;
>
> > “on which a classifier can successfully be trained on”. Do you test on an established dataset?
>
> We test on the base dataset (Table 2), which follows an established approach by domain experts of combining practical specification patterns (see Li et al.,13, page 5).
>
> &nbsp;
>
> > “The generator’s input is a real scalar random value with uniform distribution [0, 1] for each position.” It is not clear here what is mean by “each position”. I assume this is each symbol in the input? But this is not clearly explained.
>
> This would relate to each symbol in the (generator’s) output. By the nature of the Transformer architecture, we fundamentally operate on sequences. By “position”, we mean such an element of the sequence. Thanks, we will clarify this.
>
> &nbsp;
>
> > “The position-wise padding mask is copied from the real data during training, so the lengths of real and generated formulas at the same position in a batch are always identical.” The first time you refer to the padding mask it is not clear what this is or where it comes from?
>
> We refer to the standard padding mask of the Transformer, which can be computed given the allowed maximum length. For each batch of real and fake samples we compare, we want the ith sequence in both batches to have the same length. We, therefore, copy the padding mask for generated samples for both generator and critic from the respective real batch.
>
> &nbsp;
>
> > “Still, both generators are able to produce a large fraction of fully correct temporal specifications, which we find surprising” Could there be overfitting? Are the results the same across many runs? Is 0.3 a large fraction? How does this compare to generating examples randomly? This would make a good baseline (even if your models perform worse).
>
> If the reviewer refers by overfitting to duplicate instances, in Section 4.2.2 we note that out of 800K generated instances, only 0.28% were duplicates. If the reviewer means something else, we kindly ask for an elaboration.
> Results are very consistent across multiple runs. We will add further details on the variance between runs to the paper.
> Given the non-autoregressive generation, we consider 30% a large fraction.
> If the reviewer refers by “generate randomly” to sampling each position of a sequence independently at random, this would result in a very, very small fraction of correct instances, since the formula must exactly form a tree to be syntactically correct.
>
> &nbsp;
>
> > The satisfiability classifier results are interesting. What would be better tho, is to show that training on the generated data improved results on an established dataset.
>
> Testing on formulas as in the LTL base data set is the established way of testing the performance of LTL satisfiability solvers (Li et al. ‘14).
> A synthetic dataset that is designed to bring classical solvers to their limits is a portfolio dataset by Schuppan and Darmawan ‘11 of which around 750 formulas fit into our encoder token and size restrictions.
> Note that these scalable benchmarks are out-of-distribution and almost all of them are satisfiable. We nevertheless, tested two models trained on Generated and LTLbase on these benchmarks:
>
>  - Models trained on Generated tested on these benchmarks: 93.6%, 93.1% accuracy
>  - Models trained on LTLbase tested on these benchmarks: 82.1%, 93.3% accuracy

---

> ### Author Response · Authors · 2021-11-12
> **Reply to Reviewer 3zMF, part 2**
>
> > It is interesting that you can learn on data generated using the LTLbase 10k dataset and perform better on the validation set than training directly on the LRLbase (and perform similarly well when training on the whole LTLbase dataset). However, it’s clear that the model has overfit to the LTLbase 10k. How did you decide to stop training? Does this happen for all runs? It would be helpful to see the training and test curves while the model is training. Additionally, it would be good to see a graph with multiple runs.
>
> We trained each variant for an equal number of steps (30K). Validation accuracy never gets significantly higher during training for the 10K variant, though. We will add the curve to the paper.
>
> &nbsp;
>
> > For all results in the paper, it would be best to perform multiple runs and report the standard deviations.
>
> We always conducted two to three runs (three as default, otherwise indicated in the caption). The variance was small. We will include the standard deviations in the paper.
>
> &nbsp;
>
> > In the section titled “GAN with included classifier”: What are you classifying? Are you predicting satisfiability? This is not clear. How do you know the satisfiability of the generated samples? Is there a 50/50 split of satisfiable and unsatisfiable examples?
>
> Yes, we are predicting satisfiability. We used the tool aalta to label our instances, both for the base and generated dataset. Satisfiable and unsatisfiable instances are balanced 50/50 per sequence length and consequently on the whole set. Thanks, we will make this more clear in the paper.
>
> &nbsp;
>
> > Are there the same number of training examples in the LTLbase, Uncert-e and mixte-e datasets?
>
> The Generated, Mixed and Uncert datasets all contain exactly 400k instances. LTLbase contains slightly above 380K instances.
>
> &nbsp;
>
> > Table 3: Are you training and testing on different splits of the dataset? It would help to add the std? The results somewhat suggest that training to increase uncertainty does produce slightly more challenging problems, but it’s hard to tell because it’s not clear how much results vary between runs and it’s not clear if the differences are statistically significant.
>
> Yes, we are always testing on a held-out set. For this particular experiment, we did two runs with very similar results. We will run more and include the standard deviation.
>
> &nbsp;
>
> > Table 3: It’s very important to know how train and test samples were split for Uncert-e. Uncert-e being more difficult to classify could also be explained by lack of variation and the samples you tested happening to be in a different mode to those in the training set. What is the average length of problems in each data set?
>
> For Uncert-e we first sampled a large number of generated instances. Then we balanced the classes per length just like for the other sets. From this, we randomly take 400K instances for the training split. From the remaining instances, we randomly take 20K instances for the validation split.
> We will add length distributions of the generated sets to the appendix.
> The average lengths (for the train sets) are: LTLbase 34.6, Generated 33.6,  Uncert-e 38.0.
>
> &nbsp;
>
> > Is there a qualitative difference between samples generated with and without the uncertainty loss? What makes the problems harder?
>
> There are, unfortunately, no straightforward qualitative measures for the hardness of a formula, neither length nor time to solve. For example, a classical solver takes longer if the formula is unsatisfiable. We thus decided to stick to a “hardness” in terms of classification accuracy.
>
> &nbsp;
>
> > Does training on Uncert-e improve performance on LTLbase (more so than training on generated?)
>
>  -   Model trained on Uncert-e tested on LTLbase: 87.6%, 87.7% accuracy
>
> In contrast to Generated on LTLbase, where the goal is to construct formulas inside the distribution, the model trained on Uncert-e performs intendedly out-of-distribution, when being tested on LTLbase.
>
> &nbsp;
>
> > Is there a standard dataset that you can show improvement on?
>
> We, again, conducted an experiment on benchmarks from Schuppan and Darmawan ‘11, where in total 750 benchmarks fit into our model parameters. We report the results of the experiments here:
>  - Model trained on Uncert-e tested on the benchmarks: 86.7%, 84.4%
>  - Model trained on LTLbase tested on the benchmarks: 83.6%, 84.3%

---

> ### Author Response · Authors · 2021-11-12
> **Reply to Reviewer 3zMF, part 3**
>
> > Why are there not more quantitative results on the integration examples? It’s clear that a classifier can already perform very well on the LTL tasks, perhaps it would be easier to see improvements on the function integration task? It does not appear that there are any results for this in the main text?
>
> Symbolic integration is not a classification, but a complex translation task involving a Transformer decoder. It is not obvious how to apply a similar uncertainty approach. We chose the integration dataset to validate our main GAN findings for generating syntactically consistent formulas on an existing dataset in a different domain.
>
> &nbsp;
>
> > General: The paper does not clearly separate LTL results and symbolic math results. This makes some results harder to parse.
>
> Generally, for the pure generation part (no classification), we consider both domains. For everything concerned with classification, starting at Section 4.2, we only consider LTL since this poses a binary classification problem. We will clarify this earlier in the paper.
>
> &nbsp;
>
> > General: This model does not generate a supervised training dataset since you still need to use existing algorithms/ programs to compute the labels/targets. This could be a problem for datasets where the solution is intractable and would also suggest that you already have a model capable of solving the problem for which you are generating the data. What is the long term motivation of this work if you either (a) cannot generate labels/targets or (b) can already use existing algorithm to solve these problems.
>
> The long-term motivation of this line of work is the enhancement of classical algorithms for symbolic reasoning with neural reasoning engines, which can make fast predictions. Note that verifying a prediction is often much easier than computing a solution. Training neural models on symbolic reasoning problems is, however, hard, because data is sparse. Training a neural model that captures the real-world distribution as close as possible from a small data set and generates more data inside this distribution is thus very desirable.
>
> Regarding the mentioned labeling problem: Labels can be predicted by a neural model as well and checked by classical tools (models can already predict solutions to LTL when classical solvers time out; see Hahn et al. ’20). Depending on the domain and reasoning problem, we also expect it to be feasible when training only semi-supervised, e.g. with only partially labeled data.

---

> > ### Comment · Reviewer_3zMF · 2021-11-24
> > **Thanks for the response.**
> >
> > > The long-term motivation of this line of work is the enhancement of classical algorithms for symbolic reasoning with neural reasoning engines, which can make fast predictions. Note that verifying a prediction is often much easier than computing a solution.
> >
> > How would the authors create a supervised dataset? Are the authors suggesting that they would create a supervised dataset by generating a number of example problems using a GAN, (quickly) filter results using a verifier and then use solvers to find solutions? If computing solutions is hard, if this scalable?

---

> > > ### Author Response · Authors · 2021-11-25
> > > **Reply to Reviewer 3zMF**
> > >
> > > > How would the authors create a supervised dataset? Are the authors suggesting that they would create a supervised dataset by generating a number of example problems using a GAN, (quickly) filter results using a verifier and then use solvers to find solutions? If computing solutions is hard, if this scalable?
> > >
> > > Please note that this is out of the scope of this paper. But that being said, the reviewer’s question is interesting.
> > > A scalable generation of a supervised dataset works indeed as follows: 1) Generate large numbers of problem instances from the desired distribution (even from a small data source and even if no rules exist to generate these algorithmically). 2) Label (with a timeout) as many instances as possible. 3) Add labeled data to the dataset.
> > >
> > >
> > > Such datasets are very much useful:
> > >
> > > a) Despite using a timeout for the classical tool to label the data, the NN can generalize to the semantic concepts or patterns in the data, which classical solvers (usually based on searching through the state space) cannot. For example, Transformers trained on LTL tasks (Hahn et al ‘21, Schmitt et al ‘21), even solved instances where classical algorithms timed out.
> > >
> > > b) This can be done completely a priori and we expect greatly increased performance during runtime. For example, a supervised trained version of NeuroSAT was being used to predict unsat-cores to increase SAT solver performance significantly (Selsam and Bjorner ‘19).
> > >
> > > We thank the reviewer for their questions. We can add a brief discussion on this to the conclusion if the reviewer finds this interesting.

---

### Official Review · Reviewer_vhbE · 2021-11-03

**Correctness:** 3
**Technical Novelty And Significance:** 2
**Empirical Novelty And Significance:** 2
**Recommendation:** 5
**Confidence:** 3

**Main Review:**

** Update after author responses ** After the author clarified with the format of the data and results, my problems have resolved in part and thus I updated the score. My concern in the motivation of the paper still remains. The paper claims that synthetic dataset generated from rules is limited in the coverage so that it is valuable to construct synthetic dataset from neural models --- however, it is still not clear to me if experiments demonstrate that the resulting data from neural models are shown to have better coverage and higher quality than data from rules.

The model introduced in the paper is well-executed, and based on the generated data attached in the submission, the data quality appears to be good enough. There are still a few concerns I have in the motivation of the problem and experiments.

First, the problem setup is not convincing to me and is not well-motivated in the paper. If data for the symbolic reasoning --- which is not precisely defined in the paper --- can easily be generated at scale using rules, as done for the base data in the paper, what is the reason for training a neural model to generate more data? How is it inherently different from generating more using rules, as data, either generated automatically from rules or generated by neutral models, is equally artificial?

~~Second, although the paper claims that the generated data is high quality, it seems to have quite bad quality to me. I checked Supplementary materials and most formulas contain many repetitions like “&&&&&&&GXaG>!” and are very hard to interpret. It is very difficult to find good examples as listed in Section 4.2.1, so there is a high likelihood that these good examples are cherry-picked.~~

Third, the experiments in Section 4 do not demonstrate that the generated data has high quality and effectively replaces the base data. For example, Table 2 shows that training on generated data achieves performance that is comparable to the model trained on the original data, but not better. This is related to my first point in the motivation of the problem - if the original data can be obtained automatically at scale and generated data from the proposed model is not significantly better than the original data, is there a justification for not using the original data?



**Summary Of The Paper:**

This paper aims to generate training data for symbolic reasoning by training GANs based on Transformers. Specifically, the paper explores two methods --- standard GAN and Wasserstein GAN --- which has a Transformer as an encoder and a decoder, respectively. Training these models on symbolic reasoning data that is randomly generated --- LTL and Symbolic mathematics --- is found to generate high quality formulas.

**Summary Of The Review:**

The model introduced in the paper is well-executed, and based on the generated data attached in the submission, the data quality appears to be good enough. However, in my opinion, there are more fundamental issues in the motivation of the problem and whether experiments successfully justified the usefulness of the generated data from the proposed model.

---

> ### Author Response · Authors · 2021-11-12
> **Reply to Reviewer vhbE**
>
> We would like to clarify some misunderstandings.
>
> >First, the problem setup is not convincing to me and is not well-motivated in the paper. If data for the symbolic reasoning --- which is not precisely defined in the paper --- can easily be generated at scale using rules, as done for the base data in the paper, what is the reason for training a neural model to generate more data? How is it inherently different from generating more using rules, as data, either generated automatically from rules or generated by neutral models, is equally artificial?
>
> The main motivation for this research remains that symbolic reasoning problems can, in general, not be easily generated at a large scale neither randomly nor artificially by rules: 1) random generation does not reflect the distribution of real-world problem instances, which has, for example, extensively been studied for SAT (which is subsumed by LTL) and 2) artificial generation of data requires extensive domain knowledge and an understanding of the real-world distribution. Training a neural model that captures the real-world distribution as close as possible from a small data set and generates more data inside this distribution is thus very desirable and at the same time very challenging.
>
> The LTL base data set follows work by domain experts and is based on practical specification patterns. This is simply because handcrafted benchmarks are yet too few to enable stable GAN training. We do think, however, that 10K is an achievable benchmark size in the future or even already achieved in domains unfamiliar to us.
> Note that we consider the main findings of this paper to be independent of the base data sets.
>
> &nbsp;
>
> > Second, although the paper claims that the generated data is high quality, it seems to have quite bad quality to me. I checked Supplementary materials and most formulas contain many repetitions like “&&&&&&&GXaG>!” and are very hard to interpret. It is very difficult to find good examples as listed in Section 4.2.1, so there is a high likelihood that these good examples are cherry-picked.
>
> There is a misunderstanding here about the formula representation. The formulas are stored in prefix (polish) notation. Having many top-level conjunctions is very reasonable and reflects the common structure of LTL specifications. `G` represents the globally operator, `X` the next operator, `U` the until operator, `W` the weak-until operator, `F` the finally operator, and `&`, `|`, `->`, and `!` represent the boolean connectives.
>
> Assuming with the prefix `&&&&&&&GXaG>!` you refer to line 280601 of the Uncert-e dataset: The complete formula reads
>
>     GXa & G(!(g & Xa) -> !f W g) & !d & GGa & G!a & GX((g <-> !a) | GX(XXc & !a)) & FFXXh & g
>
> Addressing your further concerns, here are the first 5 formulas from our generated dataset (`generated_raw.txt`) in the supplementary material, in infix notation:
>
>     (FG!(c <-> b) -> FG!d) & !f U (Xb -> (b <-> Xb))
>     Xe & G((a -> Xd) -> a) & !X(e <-> e) W !d W i & X(Xh & Xe) & F!!(h <-> a) & Xg
>     g & XXXXXXX!i & GFg
>     !!h W !d & F!!e & F!h & FXg & G(G((c -> !e) -> e | e) -> XXX(h & !e))
>     F(G!c & !Xc W Xg) & F!Xh & XXf & !(c & c) & XXg & G((h -> c) & G!Fh -> F(h & XXh))
>
> &nbsp;
>
> > Third, the experiments in Section 4 do not demonstrate that the generated data has high quality and effectively replaces the base data. For example, my interpretation of Figure 3, Table 2, and Table 3 are that distinguishing real data and generated data is overall very easy.
>
> Figure 3 shows the critic/discriminator predictions during GAN training. How well the critic/discriminator is able to distinguish real and generated data has nothing to do with base data replacement for classification. These are two entirely different problems.
> Tables 2 and 3 assess the ability of a completely independently trained classifier on the satisfiability problem. We show both how our Generated data can be used as a meaningful substitute for only 10K base training data and how the uncertainty objectives create data that is harder to classify.

---

> ### Author Response · Authors · 2021-11-20
> **Reply to update by Reviewer vhbE**
>
> We are happy that we could resolve some of the misunderstandings. We would like to thank the reviewer for taking the time to reassess.
>
> Regarding the concerns “whether experiments successfully justified the usefulness of the generated data from the proposed model”: We would like to emphasize that the model could generate enough data (400K) to train an LTL satisfiability classifier out of only 10K base data. We think that 10K data of real-world instances is an achievable size for symbolic reasoning domains.

---

> > ### Comment · Reviewer_vhbE · 2021-11-30
> > **Thanks for additional comments**
> >
> > I appreciate authors for making clarifications and providing additional details. However, I would keep the score as it is. I would like to mention that multiple other reviewers seem to share the same concerns as me, based on their final official comments, e.g., Reviewer JuGn ("I still don't find generating syntactically-correct formulae alone to be particularly interesting, since it is easy to randomly generate well-formed formulae.") and Reviewer qNn9 ("I'm not excited about just generating syntactically correct problems because a similar task has been approached by symbolic reasoning (as in program synthesis) and even by neural methods (e.g., as the following [1]). Perhaps I might miss something, but I can't find from the paper nor response why the experimental result on symbolic integration is exciting or worth reporting.").

---

### Author Response · Authors · 2021-11-20
**Revised version**

We have uploaded a revised version of the paper. We list the changes in the following:
- A clarification in the introduction that experiments on classification and uncertainty were carried out for LTL.
- Adjusted some wordings and fixed typos as suggested by reviewer qNn9 and JuGn.
- Added related work to data augmentation with GANs for images as suggested by reviewer qNn9.
- Added training curves for the data substitution experiments to the appendix (reviewer 3zMF).
- Ran more experiment iterations and added the standard deviations to all major tables (reviewer 3zMF).
- Added size plots of the Generated and Uncert-e sets to the appendix and noted the average formula lengths (reviewer 3zMF).
- Added an experiment with uniform noise for the GAN variant to the appendix (reviewer 3zMF).
- Expanded Table 1 to include performance decrease with stronger noise (reviewer 3zMF).
- Added experiments for different numbers of shared layers for the included classifier architecture (reviewer qNn9).
- Included out-of-distribution evaluations on a synthetic benchmark set for LTLbase and Generated to the appendix: LTLbase 86.0%, Generated 94.1% (reviewer 3zMF).
- Included out-of-distribution evaluations on a synthetic benchmark set for Uncert-e to the appendix (reviewer 3zMF).
- Included proposed clarifications in the paper.

---

### Decision · Program_Chairs · 2022-01-20

**Decision:**

Reject

**Comment:**

The paper aims to improve complex reasoning. In this regard, authors identify that acquisition of data for symbolic reasoning domains is a challenge and propose generating the data by GANs. A transformer-based architecture is proposed and trained for LTL and Symbolic mathematics. Experiments show samples generated are of good quality (e.g., correct syntax). We thank the reviewers and authors for engaging in an active discussion. However, the reviewers did not find the task of such data on its own not to be particularly interesting. Also, neither the architecture nor the training algorithm is very novel. If authors could provide a complete story i.e., show the augmented data can improve the performance of neural models that compute solutions, etc., it would make the paper much stronger. Thus, unfortunately I cannot recommend an acceptance of the paper in its current form.